# Enhanced SOLOv2: An Effective Instance Segmentation Algorithm for Densely Overlapping Silkworms

**DOI:** 10.3390/s25185703

**Published:** 2025-09-12

**Authors:** Jianying Yuan, Hao Li, Chen Cheng, Zugui Liu, Sidong Wu, Dequan Guo

**Affiliations:** School of Automation, Chengdu University of Information Technology, Chengdu 610225, China; yuanjy@cuit.edu.cn (J.Y.); lh0612123@163.com (H.L.); chengc0728@163.com (C.C.); lzgcuit2@163.com (Z.L.); wsd@cuit.edu.cn (S.W.)

**Keywords:** intelligent sericulture, silkworm instance, segmentation enhanced, SOLOv2 EAMF-Net

## Abstract

Silkworm instance segmentation is crucial for individual silkworm behavior analysis and health monitoring in intelligent sericulture, as the segmentation accuracy directly influences the reliability of subsequent biological parameter estimation. In real farming environments, silkworms often exhibit high density and severe mutual occlusion, posing significant challenges for traditional instance segmentation algorithms. To address these issues, this paper proposes an enhanced SOLOv2 algorithm. Specifically, (1) in the backbone network, Linear Deformable Convolution (LDC) is incorporated to strengthen the geometric feature modeling of curved silkworms. A Haar Wavelet Downsampling (HWD) module is designed to better preserve details for partial visible targets, and an Edge-Augmented Multi-attention Fusion Network (EAMF-Net) is constructed to improve boundary discrimination in overlapping regions. (2) In the mask branch, Dynamic Upsampling (Dysample), Adaptive Spatial Feature Fusion (ASFF), and Simple Attention Module (SimAM) are integrated to refine the quality of segmentation masks. Experiments conducted on a self-built high-density silkworm dataset demonstrate that the proposed method achieves an Average Precision (AP) of 85.1%, with significant improvements over the baseline model in small- (APs: +10.2%), medium- (APm: +4.0%), and large-target (APl: +2.0%) segmentation accuracy. This effectively advances precision in dense silkworm segmentation scenarios.

## 1. Introduction

The sericulture industry, as a vital component of the agricultural economic sector, plays an indispensable role in promoting rural economic development, increasing farmers’ income, and driving progress in the textile industry in major silk-producing countries such as China and India. With the transition of modern agriculture towards intelligent and precision-oriented practices, the research and application of smart sericulture technologies have emerged as the core driving force for enhancing industrial efficiency. In this context, computer vision technology has gradually become a critical solution for silkworm growth monitoring, behavioral analysis, and health assessment due to its non-contact nature and operational efficiency. Dense silkworm instance segmentation, serving as the foundation for individual silkworm vitality monitoring, directly determines the accuracy of subsequent biological parameter estimations for each silkworm. Consequently, it constitutes a critical technical foundation for constructing intelligent sericulture systems.

Currently, in the field of intelligent sericulture, silkworm segmentation primarily involves two approaches: traditional image segmentation techniques and deep learning-based methods. For traditional approaches, threshold-based segmentation and clustering-based segmentation are commonly utilized. Threshold-based segmentation separates silkworms as foreground from the background by analyzing feature differences and calculating optimal thresholds. Yang et al. [1] used Otsu and maximum entropy to determine the segmentation threshold for silkworms and mulberry branches in an automated feeding system for the fifth-instar Bombyx mori. Hu et al. [2] proposed a segmentation method for Bombyx mori microsporidia that combines fuzzy contrast enhancement and color feature extraction. This approach first enhances the image, then segments the microsporidia using Otsu on the color channel of the microsporidia. Clustering-based algorithms partition images by grouping pixels with similar features into distinct categories. Luo et al. [3] proposed an improved K-means algorithm to address challenges in analyzing silkworm phenotypic traits, optimizing threshold computation to mitigate background interference and accurately measure silkworm areas. Although traditional methods demonstrate simplicity and computational efficiency, they are highly sensitive to illumination variations and fail to resolve overlapping individuals in dense populations.

In recent years, with advancements in deep learning, numerous researchers have applied deep learning-based image segmentation methodologies to silkworm segmentation. H et al. [4] conducted a comparative study employing three deep learning models: U-Net, Mask R-CNN, and FCN for automated silkworm segmentation. The comparative analysis demonstrated that the U-Net model achieved the highest Intersection over Union (IoU) score of 0.89 among the three approaches. Wang et al. [5] developed an automated system for Bombyx mori counting and instar stage identification by leveraging the YOLOv5 model with real-time image acquisition and instance segmentation algorithms, significantly enhancing the automation level of sericulture practices. In the related advancement, He et al. [6] introduced an improved Mask R-CNN model incorporating pixel reweighting and bounding box refinement strategies. Experimental evaluations on samples with complex background interference demonstrated that their approach achieved a silkworm recognition accuracy of 96.83%. Currently, research on silkworm instance segmentation is still in its early stages. In the realm of instance segmentation for insects similar to silkworms, they proposed an enhanced instance segmentation model combining Pyramid Vision Transformer (PVT) with Mask R-CNN, targeting the challenge of detecting camouflaged pests (e.g., cotton bollworms) [7]. This approach significantly improved pest recognition performance in complex agricultural environments. For the coffee fruit fly, Tassis et al. [8] integrated instance segmentation with semantic segmentation, achieving mean Intersection over Union (mIoU) scores of 94.25% and 93.54% for UNet and PSPNet networks, respectively. For the tomato leafminer, Uygun et al. [9] employed the YOLOv8l-Seg model for training, demonstrating that the proposed model and methodology can effectively detect damage caused by the pest. Chen et al. [10] proposed a new approach, local maximum of boundary intensity (LMBI), which generates segmentation candidates from a region growing process to achieve quantitative analysis of nematodes in soil. To identify precious tree pests in real complex natural environments, Guo et al. [11] proposed an improved instance segmentation method that replaces the backbone network of Mask R-CNN with Swin Transformer. The experimental findings demonstrated that the suggested method successfully segmented pests in a variety of situations, including shaded, overlapped, and foliage- and branch-obscured pests.

Although instance segmentation techniques have witnessed remarkable advancements in general object domains (e.g., pedestrians, vehicles), their deployment in the context of silkworm segmentation, especially under high-density breeding scenarios, has remained largely underexplored. Unlike pedestrians and vehicles, silkworm instance segmentation encounters distinct challenges stemming from the extremely high density, flexible morphological variations, and severe mutual occlusion among silkworms. To address these challenges, a dedicated silkworm dataset was first constructed, and a systematic evaluation of state-of-the-art instance segmentation methods was conducted, including YOLOv5n-seg, YOLOv5s-seg, YOLACT, SOLOv1, SOLOv2, and Mask R-CNN. Among these, SOLOv2 was identified as the most promising candidate due to its superior segmentation accuracy and was thus selected as the baseline model for further optimization. Subsequently, architectural refinements were introduced to enhance SOLOv2’s capability in handling densely overlapping silkworms. First, to overcome the limitations of the original ResNet backbone in feature extraction for dense silkworm clusters in SOLOv2, a Linear Deformable Convolution (LDConv) was incorporated, enabling better adaptation to morphological diversity. Second, conventional stride convolution was replaced with a Haar Wavelet Downsampling (HWD) module to improve feature extraction for partially visible silkworms. Third, an Edge-Augmented Multi-attention Fusion Network (EAMF-Net) was integrated between network layers to address geometric continuity issues in occluded regions. For the issue of low-quality mask generation in dense silkworm segmentation, the mask feature branch of SOLOv2 was enhanced through the sequential integration of Dynamic Upsampling (DySample), Adaptive Spatial Feature Fusion (ASFF), and Simple Attention Module (SimAM), thereby producing higher-quality mask features.

The main contributions of this paper are as follows:Dataset Construction: A dedicated dataset for silkworm segmentation was built, comprising 1264 images captured under high-density breeding conditions to encompass diverse scenarios. This dataset provides essential data support for silkworm instance segmentation research.Baseline Model Identification: A systematic comparison of mainstream instance segmentation methods (including YOLOv5n-seg, YOLOv5s-seg, YOLACT, SOLOv1, SOLOv2, and Mask R-CNN) was conducted for silkworm segmentation. Based on comprehensive evaluation metrics, SOLOv2 was identified as the optimal baseline model.Enhanced SOLOv2 Architecture: An improved SOLOv2 architecture with dual-branch modifications was proposed. In the ResNet backbone network, three modules (LDConv, HWD, and EAMF-Net) were introduced to adapt to silkworm morphological variations, enhance feature extraction for partially visible targets, and resolve geometric continuity in occluded regions. In the mask feature branch, three components (DySample, ASFF, and SimAM) were sequentially integrated for improving mask generation quality.Comprehensive Experimental Validation: Extensive experiments were conducted on the self-constructed dataset, with both qualitative and quantitative results validating the effectiveness and accuracy of the proposed algorithm for silkworm segmentation. Notably, the method demonstrates superior segmentation precision for partially visible silkworms and achieves higher boundary adherence when segmenting overlapping silkworm regions.

## 2. Materials and Methods

### 2.1. Overall Workflow

To achieve accurate instance segmentation of silkworms in high-density overlapping scenarios, this study first constructed a silkworm image dataset that is consistent with real breeding environments. A professional image acquisition system was employed to capture silkworm images, and a standardized dataset was formed by integrating data augmentation and detailed annotation. Subsequently, systematic performance testing and comparative analysis were conducted on current mainstream instance segmentation methods using this self-built dataset, leading to the selection of SOLOv2—the most suitable method for silkworm segmentation tasks—as the baseline model. Furthermore, based on the problems exposed by the baseline model in experiments, such as the missed segmentation of partially visible silkworms and inaccurate boundaries in overlapping regions, targeted improvement strategies were designed from two dimensions: feature extraction of the backbone network and quality optimization of the mask branch. Modules including Linear Deformable Convolution (LDConv), Haar Wavelet Downsampling (HWD), Edge-Augmented Multi-attention Fusion Network (EAMF-Net), Dynamic Upsampling (DySample), Adaptive Spatial Feature Fusion (ASFF), and Simple Attention Module (SimAM) were introduced to construct the enhanced SOLOv2 algorithm. Finally, through comparative experiments and visual analysis, the effectiveness of the improved model was fully verified, and its performance advantages in silkworm instance segmentation tasks were thoroughly analyzed.

### 2.2. Dataset Construction

The dataset construction process is illustrated in Figure 1. First, the silkworm images were collected and augmented. Subsequently, instance-level mask annotations were performed. Following this, the dataset was partitioned into training, testing, and validation subsets. Finally, the dataset was converted into the standard COCO dataset format compatible with MMDetection. The detailed construction process will be elaborated on in the subsequent sections.

The data collection was conducted in the silkworm rearing rooms of silkworm farmers in Lezhi County, Sichuan Province, China, as illustrated in Figure 2. A specialized image acquisition system was constructed to obtain real silkworm data, consisting of an industrial color camera (IDS uEye SE UI-3130SE. Produced by Guangzhou Hongke Electronics Technology Co., Ltd., Guangzhou, China) with 1280 × 720 pixel resolution, two linear bar light sources, an adjustable mounting apparatus, and a data processing workstation. The camera was positioned approximately 0.2 m above the specimen tray, enabling consistent capture of the silkworms at densities of about 70 individuals per frame. To obtain silkworm data under different activity states, the data acquisition process was divided into two distinct phases: pre-feeding and post-feeding periods. Given the characteristically slow locomotion of silkworms, the imaging system was configured to capture frames at 5 s intervals to mitigate redundant data accumulation caused by excessive frame rates. The entire data collection spanned a total duration of 2 h.

To enhance the model’s robustness and generalization capabilities, multiple data augmentation strategies were applied to the collected silkworm images, including brightness adjustment, noise addition, and random rotation. After obtaining the augmented silkworm images, segmentation annotation of the silkworms was performed to create the silkworm dataset. This study employed the interactive semi-automatic image annotation tool (ISAT) for silkworm annotation. Following annotation, the dataset was partitioned into training, testing, and validation sets based on quantity and image redundancy in an 8:1:1 ratio. The final constructed silkworm dataset comprises 1264 dense silkworm images, with 1012 training images, 126 test images, and 126 validation images. To ensure the representativeness of each subset, the entire dataset was uniformly shuffled before partitioning, and the split was performed randomly to avoid sample bias caused by non-random grouping.

### 2.3. Baseline Model Selection for Silkworm Instance Segmentation

Since the inception of instance segmentation by Hariharan et al. [12] in 2014, numerous algorithms have emerged. To identify the optimal algorithm for silkworm segmentation, this study systematically evaluated the mainstream single-stage (e.g., YOLACT, SOLO, SOLOv2, YOLOv5n-seg, YOLOv5s-seg) and two-stage (e.g., Mask R-CNN) instance segmentation architectures. Quantitative experiments and qualitative analyses demonstrated that SOLOv2 achieved the highest segmentation accuracy (see [13]) for experimental details). Consequently, SOLOv2 was adopted as the baseline framework.

Despite its superior accuracy, SOLOv2 exhibits limitations in real-world silkworm breeding scenarios characterized by high-density distributions, severe inter-silkworm overlapping, and frequent occlusion by mulberry leaves. Under these conditions, SOLOv2 suffers from (1) missed segmentation of partial visual silkworms, (2) over-segmentation of individual silkworms, and (3) fragmented segmentation of overlapping instances with poor boundary adherence. These deficiencies undermine the reliability of key metrics in intelligent farming systems, such as silkworm area and perimeter estimation. To address these challenges, an enhanced SOLOv2 architecture was designed.

### 2.4. Enhanced SOLOv2 Model

This section first outlines the original SOLOv2 architecture, followed by a detailed description of our proposed enhancements.

#### 2.4.1. SOLOv2 Model Overview

As illustrated in Figure 3, the SOLOv2 architecture comprises three main components: a ResNet backbone network, a Feature Pyramid Network (FPN), and an instance segmentation network [14]. The ResNet backbone is primarily responsible for extracting image features. The FPN enhances the model’s capability to detect objects at different scales. The instance segmentation network, serving as the core component of SOLOv2, performs segmentation of individual object instances in images through three parallel processing branches: the category prediction branch (Category Branch), the convolutional kernel prediction branch (Kernel Branch), and the mask feature branch (Mask Feature Branch). SOLOv2 adopts a simple yet effective single-stage instance segmentation framework that achieves high segmentation accuracy. By simultaneously predicting semantic categories and mask categories, it enables both distinction between different objects and identification of similar objects across various regions in an image. Our improvements focus on optimizing the ResNet backbone and enhancing the mask feature branch (highlighted in red in Figure 3).

#### 2.4.2. Feature-Enhanced Backbone Network

The SOLOv2 takes ResNet-50 as the backbone. On this basis, this study introduces LDConv, HWD, and EAMF-Net to enhance the feature extraction ability.

1.Linear Deformable Convolution (LDConv)

The ResNet-50 uses traditional convolution to extract target features; however, the traditional convolution suffers from limited receptive fields and rigid geometric sampling structures, making it ineffective for silkworm segmentation, where targets exhibit significant posture changes and geometric discontinuities due to occlusion.

In response to this, this paper introduces an LDConv module in the ResNet-50 to replace the original traditional convolution module. LDConv was proposed by Zhang et al. [15]; it can obtain a variety of sampling shapes. Specifically, LDConv generates the original coordinate according to the number of parameters of the convolution kernel firstly, and then learns the offset through the convolution operation and adds the offset to the initial sampling coordinates to generate a new modified coordinate. Finally, the feature of the corresponding position is extracted by interpolation and resampling, and the convolution operation is applied to output the result. Compared with standard convolution and deformable convolution, it changes the growth trend of the number of parameters into linear growth and reduces the computational overhead.

2.Haar Wavelet-Based Downsampling (HWD)

In ResNet-50, downsampling is implemented through strided convolutions. As the convolution stride increases, the step size of the convolutional kernel on the input data becomes larger, leading to two potential issues. Firstly, degraded local feature capture: The increased stride reduces the coverage density of the convolutional kernel over local regions, making it difficult for the network to capture fine-grained features. Secondly, feature sparsity: The feature density in the output feature maps decreases, leading to sparser feature distributions, which reduces the model’s sensitivity to details. These issues directly impair the model’s ability to identify small target features, ultimately degrading performance in classification and segmentation tasks.

In silkworm segmentation, significant size variations exist among silkworms, and partial visual silkworms frequently appear at image boundaries due to the field-of-view (FOV) truncation. Traditional downsampling methods progressively lose detailed information about these small targets, as the resolution of feature maps decreases across layers. To address this problem, an HWD module was introduced to replace the strided convolutions in ResNet-50. The HWD module, proposed by Xu et al. [16], utilizes the Haar wavelet transform to decompose input feature maps into low-frequency and high-frequency components. The low-frequency components retain the main structural and global semantic information, whereas the high-frequency components represent detailed features, such as edges and textures. By fusing multi-frequency information, this module preserves both the primary morphological characteristics and fine details of silkworm targets during the downsampling process. Furthermore, this module can directly replace traditional pooling or strided convolution layers, thereby enhancing the model’s adaptability to multi-scale features with minimal computational overhead.

3.Edge-Augmented Multi-attention Fusion Network (EAMF-Net)

To address the challenges of geometric discontinuity and weak edge segmentation caused by overlapping and occluded silkworms, an EAMF-Net was proposed. The EAMF-Net innovatively integrates four components: a Channel Attention Module (CAM), a Spatial Attention Module (SAM), an Edge Enhancement Module (EEM), and a Multi-attention Fusion Module (MAFM). As illustrated in Figure 4, the CAM and SAM are connected sequentially, while the EEM is incorporated in parallel through a bypass connection. Finally, the feature maps are summed and fed into the MAFM to generate the refined output.

Within this framework, the CAM models cross-channel feature dependencies to enhance selective attention across channel dimensions, enabling better association and utilization of critical information. The SAM performs spatial sensitivity analysis to strengthen feature discriminability in spatial dimensions, improving localization and recognition of geometrically discontinuous silkworms. The EEM amplifies edge contours and texture features, facilitating precise boundary detection of overlapping specimens. The MAFM synthesizes multi-attention features through learnable fusion strategies. The detailed architectures of these modules are elaborated in the following text.

CAM: As illustrated in Figure 5a, the input feature map $X^{B\times C\times H\times W}$ is processed through parallel Adaptive Average Pooling (AAP) and Adaptive Max Pooling (AMP) operations to generate two 1D channel-wise feature weight vectors: $K_{CM}^{B\times C\times 1\times 1}$ and $K_{CA}^{B\times C\times 1\times 1}$. These vectors are then summed to produce $K_{C}^{B\times C\times 1\times 1}$. A 3 × 3 1D convolutional layer is applied to fuse the weight information, followed by a Sigmoid activation function to generate the final channel attention weights. The detailed mathematical formulation is provided in Equation (1)*X*_1_ = CAM(*X*) = σ (C1D_3×3_ (*P*_max_(*X*) + *P*_avg_(*X*))) ⊛ *X*
(1)
where CAM() denotes the channel attention function; *P*_max_() represents the max pooling operation; *P*_avg_() indicates the average pooling operation; C1D_3×3_() corresponds to a 1D convolution, where the subscript specifies the kernel size; *σ*() signifies the Sigmoid activation function; and ⊛ denotes the Hadamard product operation; *X*_1_ is the output of CAM.

SAM: As shown in Figure 5c, the feature map $X_{1}^{B\times C\times H\times W}$ output by the CAM is fed into the SAM. Within SAM, the channel-wise maximum ${\;K}_{SM}^{B\times 1\times H\times W}$ and channel-wise average $K_{SA}^{B\times 1\times H\times W}$ are extracted from $X_{1}^{B\times C\times H\times W}$. These features are concatenated along the channel dimension to form $K_{S}^{B\times 2\times H\times W}$. Parallel 3 × 3 and 11 × 11 convolutional layers are applied to $K_{S}^{B\times 2\times H\times W}$, and their outputs are summed and normalized via a Sigmoid function to produce ${\;X}_{2}^{B\times C\times H\times W}$. The detailed formulation is given in Equation (2)*X*_2_ = SAM(*X*) = σ (*C*_3×3_(Ct[Ch_max_(*X*_1_),Chavg(*X*_1_)]) + *C*_11×11_(Ct[Ch_max_(*X*_1_),Ch_avg_(*X*_1_)])) ⊛ *X*_1_
(2)
where *C*_3×3_(⋅) denotes standard 2D convolution with a 3 × 3 kernel; Ch_avg_(⋅) is channel-wise average pooling; Ch_max_(⋅) represents channel-wise max pooling; and Ct[⋅] indicates channel-wise concatenation.

EEM: As depicted in Figure 5d, the core structure of this module is the Edge Enhancement (EE) operation [17]. The EE module improves the segmentation accuracy of overlapping or weak edges in silkworm images by subtracting the feature average pooling results from the original feature map. Let ${\;X}_{e}^{B\times C_{e}\times H\times W}$ denote the input feature map of the EE module, its output $X_{ee}$ is formulated in Equation (3)*X_ee_* = EE(*X_e_*) = *X_e_* + *C*_1×1_ (*X_e_* − *P*_avg_(*X_e_*)).(3)

MAFM: As illustrated in Figure 5b, the summed features of $X_{2}^{B\times C\times H\times W}$ and $X_{ee}^{B\times C\times H\times W}$ are used as the input to the MAFM, with the computational process defined in Equations (4) and (5). Within the MAFM, three parallel wavelet-based convolutional layers with varying kernel sizes are employed to fuse multi-scale features extracted from preceding modules*X*_3_ = *X_ee_* + *X*_2_(4)*X*_4_ = *C*_1×1_ (Ct [WTC_3×3_(*X*_3_), WTC_5×5_(*X*_3_), WTC_7×7_(*X*_3_)]).(5)

The proposed EAMF-Net is incorporated into the SOLOv2 backbone network after Stages 1 to 4, with its detailed architectural configuration illustrated in Figure 6. Where H and W represent the height and width of the image, respectively. During training, the image size is randomly selected from [(720, 1280), (640, 1280), (560, 1280), (480, 1280)], while a unified size of (560, 1280) is used during testing.

The EAMF-Net is integrated into the backbone network, with the specific connectivity mechanism between residual blocks of adjacent stages illustrated in Figure 7.

#### 2.4.3. Enhanced Mask Feature Branch Network

The core functionality of the SOLOv2 mask feature branch lies in constructing foundational features for instance segmentation through multi-scale feature fusion. As depicted in Figure 7, this module processes four-level feature maps from the Feature Pyramid Network (FPN). Initially, hierarchical features are unified to 1/4 of the input image resolution via convolutional operations and bilinear upsampling to achieve spatial dimension standardization. Subsequently, multi-scale information is integrated through progressive summation of feature maps, yielding a composite feature tensor for mask prediction. To enhance silkworm segmentation quality, this study augments the original SOLOv2 mask feature branch by incorporating three novel components: (1) the DySample network that replaces fixed bilinear interpolation with learnable upsampling for adaptive spatial detail recovery; (2) the ASFF network that optimizes cross-scale feature integration using channel-wise and spatial attention mechanisms; and (3) the SimAM, which strengthens feature discriminability without computational overhead. This refined architecture preserves the primary morphological integrity of silkworm targets while effectively capturing fine-grained edge and texture details, thereby improving segmentation accuracy for partially visible specimens and multi-scale objects in complex imaging scenarios.

1.Dynamic Upsampling Module (Dysample)

In Figure 7, when the summation of the four-layer feature maps (P2 to P5) is performed, upsampling is required for feature maps of varying resolutions. In the original SOLOv2 model, bilinear interpolation is employed for this operation. However, this method fails to account for the influence of grayscale gradient variations among adjacent pixels on interpolation results, which often leads to the loss of edge and texture information after upsampling. Consequently, the clarity and fine details of target objects are inadequately preserved, resulting in edge blurring. To address this limitation, the DySample upsampling module, proposed by Liu et al. at ICCV 2023 [18], is introduced. Its core principle is defined as point sampling, where content-aware sampling points are generated by learning coordinate offsets from input feature maps, enabling adaptive resampling of feature maps.

2.Adaptively Spatial Feature Fusion (ASFF)

In the original SOLOv2, the summed P2-P5 features are spatially normalized and directly aggregated to form a composite feature tensor, which is then convolved to generate the mask feature. This approach neglects semantic discrepancies across hierarchical feature levels during fusion, which can induce cross-target feature confusion and geometric distortions in instance mask shapes. To mitigate this, an ASFF module [19] is integrated. By leveraging adaptive pooling, secondary weight learning for semantic information across different hierarchical levels is facilitated, thereby enhancing the fusion of local features and improving segmentation accuracy under occlusion scenarios.

3.Simple Attention Module (SimAM)

The SimAM, a lightweight and parameter-free attention module, is designed to emphasize feature saliency by suppressing activations of neighboring neurons [20]. This module is demonstrated to significantly boost recognition accuracy in dense scenes for object detection and enhance boundary sensitivity in image segmentation models. Unlike conventional channel or spatial attention mechanisms, 3D attention weights for each neuron are inferred by computing feature similarity through SimAM, thereby amplifying critical features while suppressing irrelevant ones without introducing additional parameters to the baseline network. In this work, the SimAM module is appended to the final layer of the segmentation head, following the output of the ASFF module.

The improved mask feature branch is illustrated in Figure 8, where components highlighted in red are defined as the novel modules introduced in this study. Where H and W represent the height and width of the image, respectively. During training, the image size is randomly selected from [(720, 1280), (640, 1280), (560, 1280), (480, 1280)], while a unified size of (560, 1280] is used during testing.

## 3. Experimental Results and Discussion

### 3.1. Experimental Setup

The experimental hardware configuration was configured as follows: an Intel Core 13900KF processor, an NVIDIA GeForce RTX 3090 GPU, and a Windows 10 operating system. The software environment was implemented using the PyTorch 1.10.0 deep learning framework, CUDA 11.3 for computational acceleration, Python 3.8.10, and the MMdetection 3.0.0 framework. During the model training phase, hyperparameters were set as follows: the initial learning rate was fixed at 0.00125 to balance training stability and convergence acceleration; a momentum factor of 0.9 was applied to optimize convergence and suppress oscillations; and a weight decay coefficient of 0.0001 was adopted to prevent overfitting. The model was trained for 36 epochs with a batch size of 1 using the stochastic gradient descent (SGD) optimizer. Input images were resized to 1280 × 720 pixels to balance computational efficiency and detail preservation. Evaluation results were recorded, and weight files were saved once per epoch.

### 3.2. Results Analysis

To validate the performance of the proposed algorithm in silkworm segmentation, comparative experiments were designed as follows: (1) comparison between the proposed algorithm and the original SOLOv2; (2) performance comparison with mainstream instance segmentation algorithms.

#### 3.2.1. Silkworm Segmentation Metrics

For silkworm segmentation in this paper, the IoU is first calculated, defined in Equation (6), where Mpred represents the predicted mask and Mgt represents the ground-truth mask(6)$$\mathrm{IoU}=\frac{M_{pred}\;\cap \;M_{gt}\;}{M_{pred}\;\cup \;M_{gt}}.$$

In the MMdetection framework, by setting an IoU threshold, the precision and recall of segmentation can be calculated, as defined in Equations (7) and (8). Here, TP (true positive) indicates that positive samples are correctly identified; TN (true negative) means that negative samples are accurately identified; FP (false positive) represents that negative samples are incorrectly classified as positive; and FN (false negative) denotes that positive samples are missed or misclassified(7)$$\mathrm{Precision}=\frac{TP}{TP+FP}$$(8)$$\mathrm{Recall}=\frac{TP}{TP+FN}.$$

In this paper, the mean Average Precision (mAP) is used as the core performance metric. It is obtained by calculating the Average Precision (AP) for each detection category and then taking the arithmetic mean of these AP values. Its mathematical definition is shown in Equation (9)(9)$$\mathrm{AP}=\int_{0}^{1}P(R)dR$$
where *P* represents precision and R represents recall. Let *N* denote the total number of categories, then, the mAP can be expressed as Equation (10)(10)$$\mathrm{mAP}=\frac{\sum_{i=1}^{N}{AP}_{i}}{N}.$$

In this paper, since the silkworm is the only object that needs to be segmented (*N* = 1), the AP and mAP are completely identical. Given that the dataset created in this paper follows the standard format of the COCO dataset, when training and testing instance segmentation models on the COCO dataset, common evaluation metrics include Segm_AP, Segm_AP_50_, Segm_AP_75_, Segm_AP_s_, Segm_AP_m_, and Segm_AP_l_. Among them, Segm_AP represents the average precision within the IoU threshold range from 0.5 to 0.95. Segm_AP_50_ and Segm_AP_75_ represent the segmentation precision at IoU = 0.5 and IoU = 0.75, respectively. Segm_AP_s_, Segm_AP_m_, and Segm_AP_l_ represent the segmentation precision for small objects (area less than 32^2^ px), medium objects (area between 32^2^ and 96^2^ px), and large objects (area greater than 96^2^ px), respectively. This paper selects Segm_AP, Segm_AP_s_, Segm_AP_m_, and Segm_AP_l_ to measure the segmentation performance of the proposed algorithm.

#### 3.2.2. Ablation Study on Improved SOLOv2

To investigate the impact of different modules on the segmentation performance of the SOLOv2 network, ablation experiments were conducted by incrementally adding proposed modules to the baseline SOLOv2. In the table, “√” indicates the inclusion of a specific module, while “×” denotes its exclusion. The model size and instance segmentation metrics (AP, AP_s_, AP_m_, AP_l_) are compared in Table 1.

As shown in Table 1, the AP value was progressively improved by sequentially integrating LDConv, HWD, DySample, adaptive spatial fusion (ASFF), SimAM, and multi-attention fusion (EAMF-Net) into SOLOv2. The AP value was enhanced by 0.8%, 1.1%, 0.5%, 0.7%, 0.3%, and 0.8%, respectively, relative to the previous configuration. Overall, the AP value was increased by 4.2% in the final improved model. The segmentation accuracy for small-, medium-, and large-scale silkworm targets was significantly improved by 10.2%, 4.0%, and 2.0%, respectively. Notably, the HWD, EAMF-Net, and SimAM modules contributed to 3.1%, 2.1%, and 2.0% improvements in small-target segmentation accuracy. The model size was increased from 178 MB to 306 MB. The largest parameter expansion (97 MB) was observed when the EAMF-Net module was integrated into the backbone network. Conversely, replacing traditional convolutions with LDConv reduced the parameter size by 18 MB while improving segmentation performance.

#### 3.2.3. Comparative Experiments with Mainstream Instance Segmentation Methods

To comprehensively evaluate the performance of the improved algorithm, comparative experiments were conducted between the enhanced SOLOv2 model and several high-performance instance segmentation algorithms, followed by detailed analysis. As instance segmentation represents a critical phase in image processing research, with its primary research period spanning from 2017 to 2021 before being largely superseded by panoptic segmentation tasks, this study selected two-stage instance segmentation algorithms—Mask Scoring R-CNN [21] and Cascade Mask R-CNN [22] and the single-stage algorithm CondInst [23]—as benchmarks. The former two models, as derivatives of the Mask R-CNN architecture, have long been recognized for their high accuracy, while CondInst also exhibits robust performance. The comparative results are presented in Figure 9.

As shown in Table 2, the proposed enhanced SOLOv2 achieves the highest overall average precision (AP: 0.851) among evaluated instance segmentation models, outperforming both two-stage methods (Mask Scoring R-CNN: 0.794; Cascade Mask R-CNN: 0.721) and single-stage counterparts (original SOLOv2: 0.809; CondInst: 0.814). Notably, it demonstrates substantial improvements in small-target segmentation (*A**P*_*s*_: 0.728 vs. original SOLOv2’s 0.626), while maintaining leading performance on medium (*A**P*_*m*_: 0.858) and large objects (*A**P*_*l*_: 0.876). Despite a model size increase to 306 MB (72% larger than the original SOLOv2), it remains more compact than two-stage architectures (464–564 MB) and achieves real-time inference at 105.4 ms, balancing accuracy and efficiency for multi-scale segmentation tasks.

#### 3.2.4. Visual Evaluation

Figure 9 presents a comparative case of the segmentation results between the SOLOv2 algorithm and the proposed method for silkworm instance segmentation. In Figure 9a, the blue rectangular boxes highlight various failure cases of the original SOLOv2 algorithm, including missed segmentation in overlapping silkworm regions (Region 1), missed segmentation of partially visible silkworms (Region 2), over-segmentation (Region 3), and inaccurate boundary segmentation where the predicted edges deviate from the true boundaries (Region 4). In Figure 9b, the method proposed in this paper addresses all the aforementioned issues. Figure 10 provides zoomed-in comparisons for these four cases.

In Figure 10a, comparing the left (SOLOv2) and right (proposed method) results reveals that the original SOLOv2 algorithm fails to segment overlapping silkworms, whereas our approach achieves accurate segmentation. Figure 10b shows a silkworm (bottom-left) with only minimal head visibility. While SOLOv2 fails to segment it, our method successfully detects and segments the silkworm. In Figure 10c, two overlapping silkworms are displayed, with their segmentation masks represented by different colors. The SOLOv2 result incorrectly assigns multiple colors to each silkworm (indicating over-segmentation), while our method generates a single, consistent mask for each silkworm. Finally, Figure 10d demonstrates that SOLOv2 produces a significant boundary misalignment at overlapping regions (left), whereas our method achieves highly precise boundary adherence (right).

Figure 11 presents additional segmentation examples. Due to space limitations, only the full-image segmentation results are displayed here, without zoomed-in comparative analysis. However, it is evident that the segmentation inaccuracies present in Figure 11a,c are effectively resolved in Figure 11b,d.

#### 3.2.5. Segmentation Confidence

The confidence of all instances in the images from the experimental results was statistically analyzed. As shown in Table 3, after model refinement, our proposed method demonstrates significant improvements in confidence distribution compared to the original model. Specifically for SOLOv2, only 9.61% of instances achieved high confidence (score > 0.9), whereas our approach substantially increased this proportion to 88.8%. Conversely, the percentage of low-confidence instances (score < 0.7) decreased dramatically from 19.57% in the baseline to merely 1.50% in our enhanced method.

### 3.3. Feasibility Analysis for Deployment on Low-Resource Devices

The memory capacity of low-resource devices is usually limited (e.g., the Jetson Nano has 4 GB of memory, and embedded terminals typically have 1–2 GB of memory), so the model’s parameter scale and runtime memory usage are core constraints. As shown in Table 2, the model size of enhanced SOLOv2 is 306 MB. Although this is a 72% increase compared to the original SOLOv2 (178 MB), it is significantly smaller than that of two-stage models (Mask Scoring R-CNN: 464 MB; Cascade Mask R-CNN: 564 MB) and far below the memory upper limit of low-resource devices (easily loadable on devices with 4 GB of memory).

In terms of runtime memory usage, the memory consumption of the model during the inference phase mainly comes from feature map storage. Based on the tensor dimension design in Section 2.4.2 and Section 2.4.3, the maximum dimension of the feature map output by the backbone network is B × 512 × 80 × 45 (approximately 1.8 MB when B = 1), and the maximum dimension of the feature map in the mask branch is B × 256 × 320 × 180 (approximately 18.4 MB when B = 1). The total feature map memory usage for single-sample inference is less than 50 MB, which can meet the requirements even on embedded devices with 1 GB of memory.

In addition, the model supports dynamic batch inference (inference latency of 105.4 ms when B = 1, Table 2). The memory pressure can be further reduced by controlling the batch size, enabling adaptation to low-memory devices. Therefore, the model proposed in this paper can be deployed on low-resource devices.

## 4. Conclusions

Instance segmentation of silkworms is the foundation for achieving intelligent sericulture; however, there is currently limited research in this area. This paper explores instance segmentation of silkworms. Firstly, a dataset specifically for silkworm segmentation was self-constructed, and current mainstream instance segmentation algorithms (Mask R-CNN, YOLACT, YOLO v5s, YOLO v5n, SOLOv1, and SOLOv2) were tested on this dataset. The SOLOv2 algorithm, which achieved the highest segmentation accuracy, was selected as the baseline model for silkworm segmentation. Subsequently, to address the issues of missed segmentation of incomplete silkworms and mis-segmentation of overlapping silkworms that still persisted when using SOLOv2 for silkworm segmentation tasks, an enhanced SOLOv2 model tailored for silkworm segmentation was designed. Specifically, to tackle the problem of the ResNet backbone network’s insufficient feature extraction capability for dense silkworms, LDConv was introduced into the ResNet backbone network to better adapt to the variable morphologies of silkworms. Traditional strided convolution was replaced with an HWD to enhance the feature extraction capability for small, incomplete silkworm targets. An edge-enhanced multi-attention fusion network was incorporated between stages to better parse the geometric continuity of occluded silkworms. Additionally, to address the low-quality mask generation for dense silkworm segmentation in the mask feature branch network, DySample, ASFF, and the SimAM parameter-free attention module were sequentially introduced to generate higher-quality mask features.

The experimental results demonstrate that the proposed algorithm in this paper improves upon the original SOLOv2 by 4.2%, 10.2%, 4.0%, and 2.0% in terms of the AP, APs, APm, and APl metrics, respectively. Visualization of the segmentation results reveals that the proposed algorithm enhances the segmentation performance for small, incomplete silkworm targets and overlapping silkworms. Meanwhile, to fully demonstrate the superiority of the improved algorithm, comparative experiments were also conducted with Mask Scoring R-CNN, Cascade Mask R-CNN, and CondInst. The improved model proposed in this paper also achieved certain improvements in the AP, AP_s_, and AP_l_ metrics. In addition to silkworm segmentation, the method proposed in this paper can also provide a reference for the instance segmentation of other agricultural pests such as wheat aphids and orchard red spiders.

## Figures and Tables

**Figure 1 sensors-25-05703-f001:**
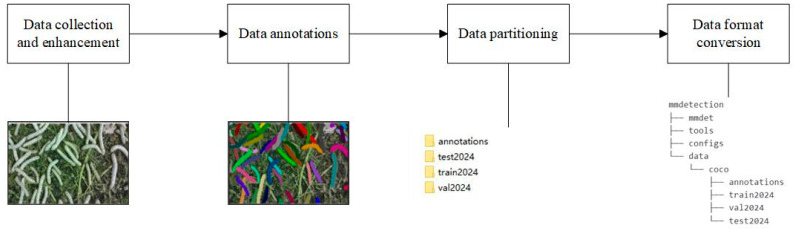
Silkworm data construction.

**Figure 2 sensors-25-05703-f002:**
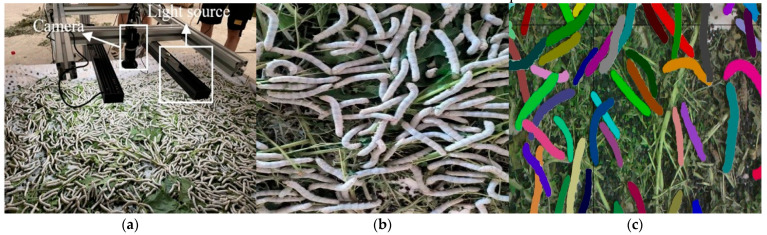
Data acquisition: (**a**) Hardware; (**b**) Silkworm image; (**c**) Annotated image.

**Figure 3 sensors-25-05703-f003:**
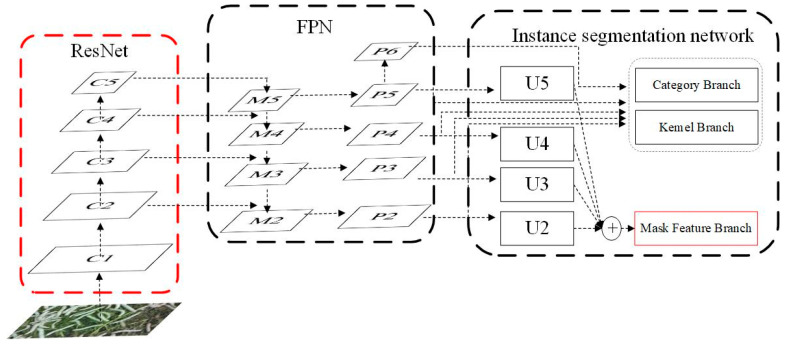
The architecture of Solov2.

**Figure 4 sensors-25-05703-f004:**
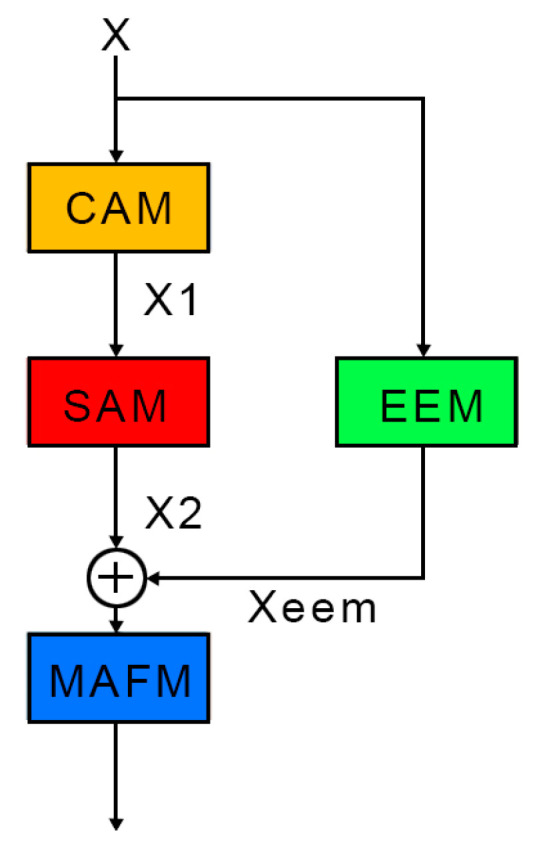
Structural Diagram of the EAMF-Net Module.

**Figure 5 sensors-25-05703-f005:**
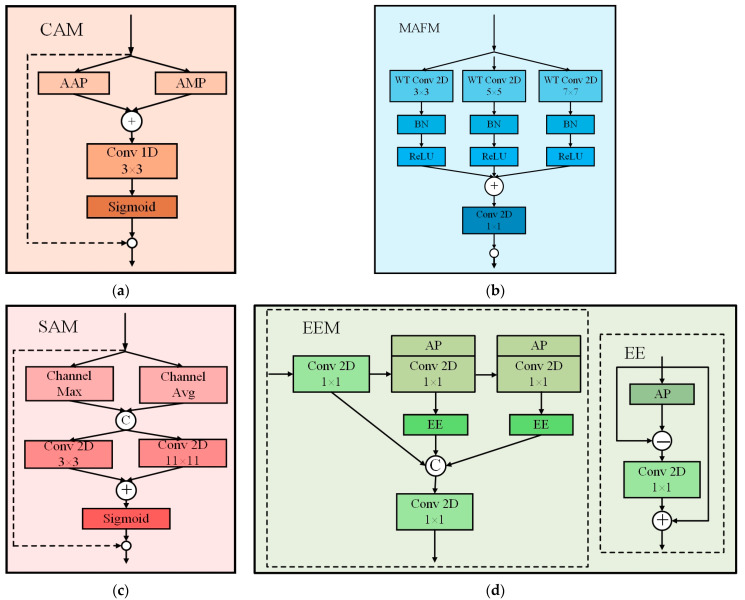
Block diagrams of the structures of each module: (**a**) CAM; (**b**) MAFM; (**c**) SAM; (**d**) EEM.

**Figure 6 sensors-25-05703-f006:**
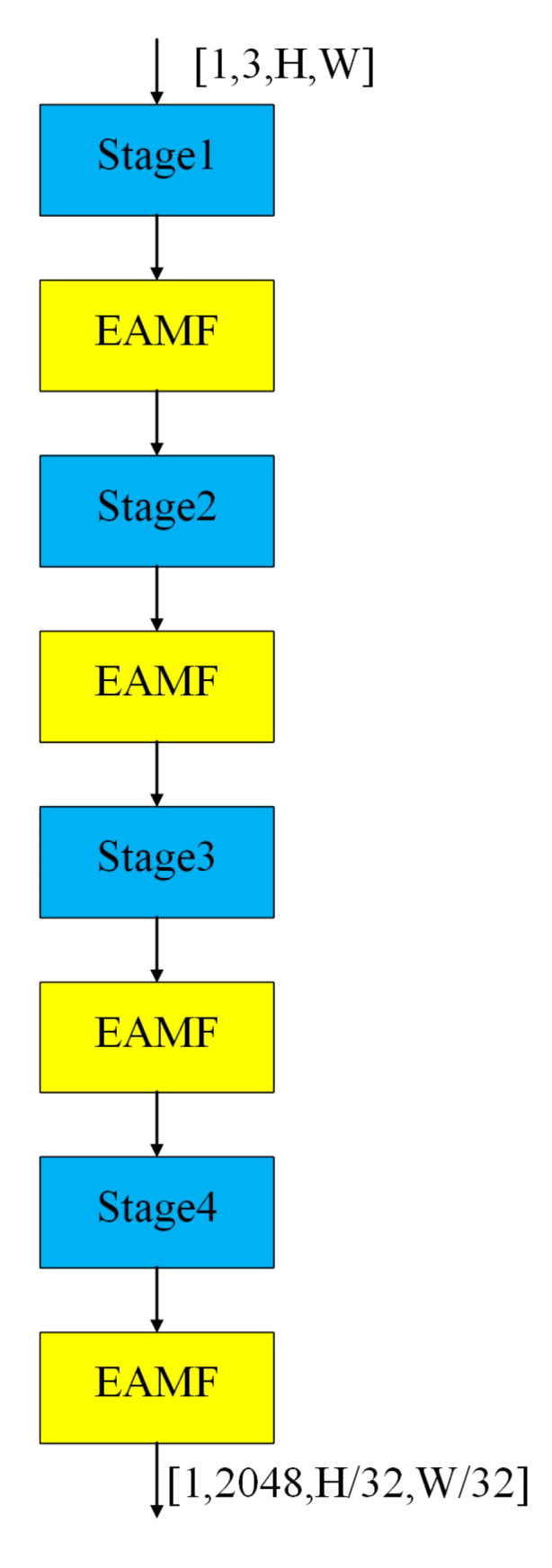
Architecture diagram of the backbone network with the integrated EAMF-Net.

**Figure 7 sensors-25-05703-f007:**
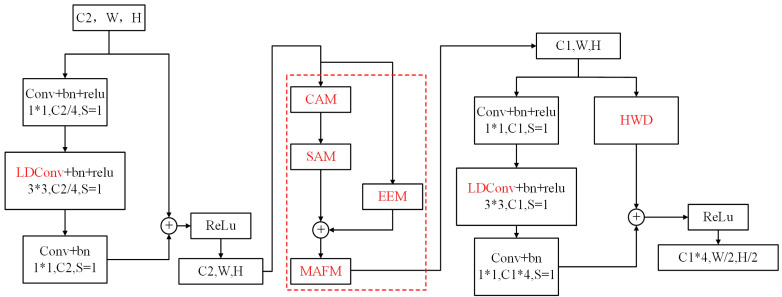
Schematic diagram of the connection between the EAMF-Net module and the residual blocks in the preceding and succeeding stages.

**Figure 8 sensors-25-05703-f008:**
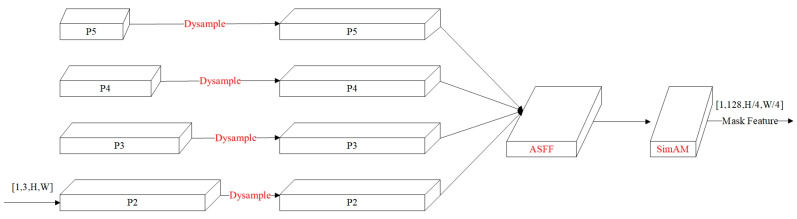
Enhanced mask feature branch network.

**Figure 9 sensors-25-05703-f009:**
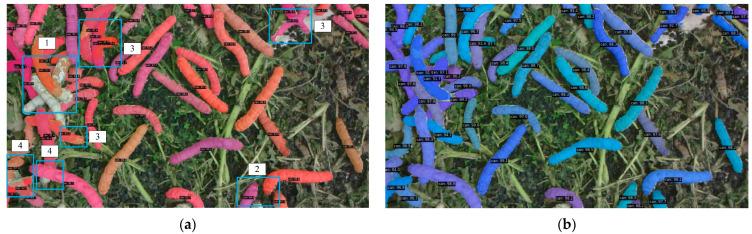
The segmentation results of two methods: (**a**) result of SOLOv2; (**b**) result of our method.

**Figure 10 sensors-25-05703-f010:**
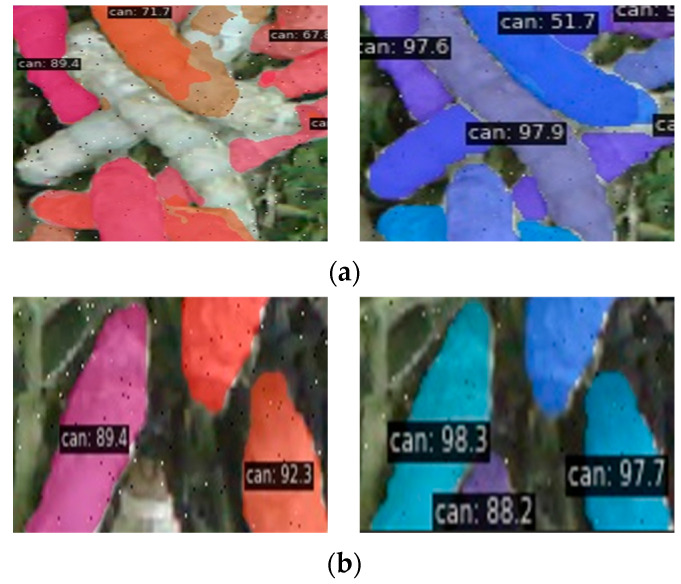
Zoom in on areas 1 to 4 in Figure 9: (**a**) zoom in on area 1 (left: SOLOv2, right: our method); (**b**) zoom in on area 2 (left: SOLOv2, right: our method); (**c**) zoom in on area 3 (left: SOLOv2, right: our method); (**d**) zoom in on area 4 (left: SOLOv2, right: our method).

**Figure 11 sensors-25-05703-f011:**
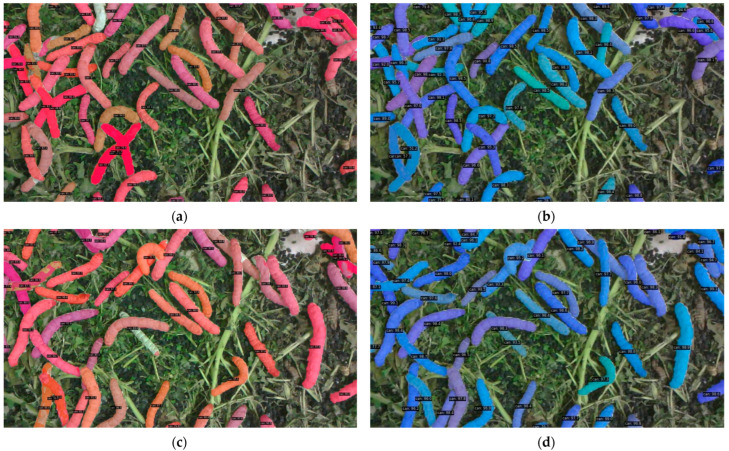
Segmentation examples: (**a**) result of SOLOv2; (**b**) result of our method; (**c**) result of SOLOv2; (**d**) result of our method.

**Table 1 sensors-25-05703-t001:** Ablation analysis of key components in dense silkworm segmentation.

Module	SOLOv2						AP	AP_s_	AP_m_	AP_l_	Size (Mb)
Baseline	×	×	×	×	×	×	0.809	0.626	0.818	0.856	178
LDConv	√	×	×	×	×	×	0.817	0.638	0.823	0.867	160
HWD	√	√	×	×	×	×	0.828	0.669	0.837	0.872	208
Dysample	√	√	√	×	×	×	0.833	0.682	0.848	0.871	209
ASFF	√	√	√	√	×	×	0.840	0.687	0.847	0.876	209
SimAM	√	√	√	√	√	×	0.843	0.707	0.851	0.869	209
EAMF-Net	√	√	√	√	√	√	0.851	0.728	0.858	0.876	306

**Table 2 sensors-25-05703-t002:** Performance comparison of different models.

Category	Model	AP	AP_S_	AP_m_	AP_l_	Size (Mb)	Time (ms)
Two-stage	Mask Scoring R-CNN	0.794	0.753	0.786	0.834	464	120.8
	Cascade Mask R-CNN	0.721	0.734	0.704	0.791	564	140.2
Single-stage	SOLOv2	0.809	0.626	0.818	0.856	178	88.3
	CondInst	0.814	0.661	0.818	0.871	236	100.5
Ours		0.851	0.728	0.858	0.876	306	105.4

**Table 3 sensors-25-05703-t003:** Segmentation results with different confidence levels (high/medium/low).

Confidence Levels	SOLOv2 (%)	Ours (%)
Highly confident (>0.9)	9.61	88.80
Confident (0.7–0.9)	78.83	9.70
Low confidence (<0.7)	19.57	1.50

## Data Availability

The raw data supporting the conclusions of this article will be made available by the authors on request.

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
