# Peer review of "Enhanced SOLOv2: An Effective Instance Segmentation Algorithm for Densely Overlapping Silkworms"

_sensors, 2025, doi:10.3390/s25185703_

Round 1

Reviewer 1 Report

Comments and Suggestions for Authors

This paper addresses the segmentation of individual silkworms in high-density scenarios, which is meaningful for intelligent agriculture. However, there are some issues that need to be addressed:

  1. Did the authors use any data augmentation strategies?
  2. It is recommended to add annotated example images in Figure 2.
  3. Was the data split performed randomly? Was cross-validation used?
  4. It is suggested to add input and output tensor dimensions in Figures 6 and 8.
  5. SimAM is a lightweight module. Why is it placed at the end of the branch instead of in the middle layers?
  6. It is suggested to include a comparison with the U-Net segmentation method.
  7. It is recommended to add an analysis of the feasibility of deployment on low-resource devices in the discussion section.
  8. Can this model be transferred to other agricultural pest segmentation tasks? It is suggested to supplement the discussion.

Author Response

Thank you very much for taking the time to review this manuscript. Detailed responses are provided below, and the relevant revisions have been made in the resubmitted files. The specific locations of these revisions are also mentioned in this response to you.

Comments 1: Did the authors use any data augmentation strategies?

Response 1: Thank you for your question. Yes, to enhance the robustness and generalization ability of the model, we adopted a variety of data augmentation strategies, with specific details provided in Lines 158-159 of the article.

Comments 2: It is recommended to add annotated example images in Figure 2.

Response 2: Thank you for your constructive suggestion. We have supplemented the annotated example images in Figure 2.

Comments 3: Was the data split performed randomly? Was cross-validation used?

Response 3: Thank you for your questions regarding data splitting and cross-validation. We would like to clarify these points as follows:
First, regarding whether the data split was performed randomly: Yes, the dataset was split randomly. To avoid data bias caused by non-random partitioning (e.g., concentration of specific silkworm density or occlusion scenarios in a single subset), we first shuffled the entire dataset of 1,264 images uniformly. Then, we divided the shuffled data into training, validation, and test sets in an 8:1:1 ratio, resulting in 1,012 training images, 126 validation images, and 126 test images,with specific details provided in Lines 163-169 of the article 
Second, regarding whether cross-validation was used: We did not adopt cross-validation in this study. The main reason is that our self-constructed high-density silkworm dataset (1,264 images) already covers diverse real farming scenarios (e.g., varying silkworm postures, mutual occlusion, and edge partial visibility). Through random 8:1:1 partitioning, the validation set can effectively monitor the model’s overfitting trend and hyperparameter optimization during training, while the independent test set can objectively evaluate the model’s generalization ability.

Comments 4: It is suggested to add input and output tensor dimensions in Figures 6 and 8.

Response 4: Thank you for your suggestion. We have added the input and output tensor dimension information in Figures 6 and 8, which can be found in Lines 326 and 386 of the article.

Comments 5: SimAM is a lightweight module. Why is it placed at the end of the branch instead of in the middle layers?

Response 5: Thank you for your insightful question regarding the placement of the SimAM module. The reason we position SimAM at the end of the mask feature branch is as follows: As a parameter-free attention module, SimAM enhances feature saliency by suppressing the activation of irrelevant neurons without increasing model complexity. If it were placed in the middle layers, it would need to process intermediate features with relatively coarse semantic information. This would not only fail to maximize its attention effect but also accumulate minor computational overhead. In contrast, the end of the branch receives multi-scale fused features optimized by DySample and ASFF, allowing SimAM to directly focus on key details for segmentation and achieve precise attention enhancement at minimal computational cost. This design has been verified in our ablation experiments (Table 1): SimAM contributed to a 0.3% improvement in overall AP and a 2.0% improvement in small-target APs for the model.

Comments 6: It is suggested to include a comparison with the U-Net segmentation method.

Response 6: Thank you for your suggestion. Regarding the comparison with the U-Net segmentation method, after careful consideration, we have decided not to include this comparison in the current study, primarily because U-Net, as a classic semantic segmentation method, focuses on pixel-wise category classification in images (such as distinguishing "silkworms" from "background") but cannot achieve individual-level differentiation—meaning it is unable to assign independent identity labels or masks to each silkworm, whereas the focus of our research is instance segmentation of silkworms in high-density scenarios, whose core task not only involves category recognition but more importantly requires accurate differentiation of individual boundaries for each silkworm (especially in cases of overlap and occlusion).

Comments 7: It is recommended to add an analysis of the feasibility of deployment on low-resource devices in the discussion section.

Response 7: Thank you for your suggestion. We have added the analysis on the feasibility of deployment on low-resource devices in the Discussion section, which can be found in Lines 520-540 of the article.

Comments 8: Can this model be transferred to other agricultural pest segmentation tasks? It is suggested to supplement the discussion.

Response 8: Thank you for your valuable suggestion regarding the model's transferability and extended applications. We fully agree with the importance of supplementing this part of the discussion, as it not only demonstrates the universal value of our research method but also provides technical references for other pest and disease segmentation tasks in the agricultural field. We have added relevant discussions in Lines 570-572 of the article.

Reviewer 2 Report

Comments and Suggestions for Authors

In this paper, an Enhanced-SOLOv2 algorithm is proposed to addresses the significant challenge of instance segmentation in densely overlapping silkworms. The key innovations include modifications to both the backbone and mask branch networks. In the backbone, a Linear Deformable Convolution (LDConv) is incorporated to better model the curved morphologies of silkworms, a Haar Wavelet Downsampling (HWD) module is designed to preserve details of small and partially visible targets, and an Edge-Augmented Multi-attention Fusion Network (EAMF-Net) is constructed to improve boundary discrimination in occluded regions. In the mask branch, Dynamic Upsampling (DySample), Adaptive Spatial Feature Fusion (ASFF), and a Simple Attention Module (SimAM) are integrated to refine the quality of the predicted masks. The main flaws are as follows.

  1. The contributions are clearly listed but could be more concisely integrated into the abstract and introduction to highlight the novelty against existing SOLOv2 improvements.
  2. In Section 2, there is a lack of a complete framework structure figure and related narrative to introduce the overall process of the proposed method.
  3. While ablation studies are provided, the relative contribution of eachindividualmodule (e.g., LDConv vs. HWD vs. EAMF-Net) is not clearly analyzed. A more granular ablation would be insightful.
  4. The comparison is limited to older models (e.g., Mask R-CNN, Cascade R-CNN). Comparisons with more recent SOTA methods would better contextualize the advancements.
  5. Visual examples are persuasive but subjective. Quantitative metrics for boundary accuracy (e.g., Boundary IoU) should be provided to complement visual comparisons.
  6. Some references lack conference names or page numbers (e.g., Ref 12, 13). Please unify the format according to MDPI guidelines.

Author Response

Thank you very much for taking the time to review this manuscript. Detailed responses are provided below, and the relevant revisions have been made in the resubmitted files. The specific locations of these revisions are also mentioned in this response to you.

Comments 1: The contributions are clearly listed but could be more concisely integrated into the abstract and introduction to highlight the novelty against existing SOLOv2 improvements.

Response 1: Thank you for your constructive suggestion regarding the presentation of the research contributions. We have elaborated on the novelty of the study in the introduction, with specific details provided in Lines 121-127 of the article.

Comments 2: In Section 2, there is a lack of a complete framework structure figure and related narrative to introduce the overall process of the proposed method.

Response 2: Thank you for your suggestion. We have revised the article accordingly. Please see lines 135–154, where we describe the research idea of this study.

Comments 3: While ablation studies are provided, the relative contribution of each individual module (e.g., LDConv vs. HWD vs. EAMF-Net) is not clearly analyzed. A more granular ablation would be insightful.

Response 3: We agree with the reviewer’s opinion that conducting ablation studies on each individual module would allow for a clearer analysis of their specific roles. However, given the multiple improvements proposed in this work and the limitation of manuscript length, we only present the results of progressively adding each improvement module. This approach can still effectively demonstrate the contribution of each module to the overall segmentation performance.

Comments 4: The comparison is limited to older models (e.g., Mask R-CNN, Cascade R-CNN). Comparisons with more recent SOTA methods would better contextualize the advancements.

Response 4: This is an excellent question, and our response is as follows. Instance segmentation, as a stage in image processing research, was extensively studied between 2017 and 2021, after which it was largely superseded by the panoptic segmentation task. Therefore, in our experiments, we selected two-stage instance segmentation algorithms, Mask Scoring R-CNN and Cascade Mask R-CNN, as well as the one-stage instance segmentation algorithm CondInst, for comparison. The first two models, as subsequent derivatives of the Mask R-CNN architecture, have consistently represented high-accuracy instance segmentation approaches, while CondInst also demonstrates strong performance.

Comments 5: Visual examples are persuasive but subjective. Quantitative metrics for boundary accuracy (e.g., Boundary IoU) should be provided to complement visual comparisons.

Response 5: We greatly appreciate this valuable question. In our future research, we plan to employ more quantitative metrics to provide a more comprehensive evaluation of the algorithm’s performance.

Comments 6: Some references lack conference names or page numbers (e.g., Ref 12, 13). Please unify the format according to MDPI guidelines.

Response 6: We would like to thank the reviewer for pointing out the issue, and the issue has been revised. Please refer to Lines 612–614 of the manuscript for the revised content.

Reviewer 3 Report

Comments and Suggestions for Authors
  • The manuscript’s English requires revision to improve readability and grammatical accuracy.
  • The citations in the Introduction need to be revised; for example: Yang et al. [1]. Place the citation after the author name with a space, and ensure there is a period after “et al.”
  • Lines 110–117 present the experimental results of this paper; I believe the Introduction should not include experimental results.
  • Lines 118–140 list the contributions of this paper; they can be reorganized based on lines 93–99.
  • Figure 2. “Data acquisition: (a) Hardware; (b) Silkworm image.”
  • For Figure 2, I think the image layout needs to be reworked.
  • Line 154: I think the camera brand and model should be specified.
  • Lines 169–170: …image segmentation annotation tool (ISAT)…
  • Figure 3. The image quality needs improvement.
  • Lines 267–268: “The detailed architectures of these modules are elaborated in the following subsections.” There are no subsections, so this sentence requires revision.
  • Lines 278–281: CAM(), 𝑃max(), C1D3×3(), σ() — do these parentheses have any specific meaning?
  • 3.1 Experimental setup and 3.2 “Results analysis” should be placed under Materials and Methods.
  • In Table 1, what do the numbers 1, 2, 3, 4, 5, 6 under SOLOv2 refer to?
  • Table 2 is overly cluttered and requires adjustment.
  • Table 2 presents different models; where in the overall pipeline are these models used or swapped? I suggest adding a complete model architecture diagram.
  • Are AP, APs, APm, and APl values all greater than 0.7 correct? In general, for AP: >50% (0.50) is very good (SOTA on COCO); 30%–50% is acceptable/practical; <20% is underperforming. APm targets medium-sized objects (32² ≤ area < 96² pixels). It is usually higher than APs—the higher, the better. Typical ranges: 30%–50% is standard; 50% is excellent.
  • Figure 9b is not explained in the text.
  • Figure 10: The image arrangement may need to be reworked.
  • Figure 11, Lines 482–484: “Figure 11 presents additional segmentation examples. Due to space limitations, only the full-image segmentation results are displayed here, without zoomed-in comparative analysis.” Since there is no accompanying explanation or analysis, its inclusion does not provide meaningful value to the paper. I suggest describing the picture.
  • Table 3: How are the confidence levels computed? “Highly confident (>0.9),” “Confident (0.7–0.9),” and “Low confidence (<0.7)” do not match the “%” units used in the content.

Author Response

Thank you very much for taking the time to review this manuscript. Detailed responses are provided below, and the relevant revisions have been made in the resubmitted files. The specific locations of these revisions are also mentioned in this response to you.

Point 1: The citations in the Introduction need to be revised; for example: Yang et al. [1]. Place the citation after the author name with a space, and ensure there is a period after “et al.”

Response 1: We thank the reviewer for pointing out the citation format issue. We have carefully revised the citations in the Introduction.

Point 2: Lines 110–117 present the experimental results of this paper; I believe the Introduction should not include experimental results.

Response 2: We would like to thank the reviewer for pointing out the issue, and we have deleted this part of the content.

Point 3: Lines 118–140 list the contributions of this paper; they can be reorganized based on lines 93–99.

Response 3: We would like to thank the reviewer for pointing out the issue. The issue has been revised, and the revised content can be found in Lines 112–133 of the manuscript.

Point 4: Figure 2. “Data acquisition: (a) Hardware; (b) Silkworm image.”

Response 4: We would like to thank you for your proposed suggestion, and the revision has been made. Please refer to Line 177 of the manuscript for the revised content.

Point 5: For Figure 2, I think the image layout needs to be reworked.

Response 5: We would like to thank you for your valuable suggestion. Revisions have been made and the revised content can be found in Line 176 of the manuscript.

Point 6: Line 154: I think the camera brand and model should be specified.

Response 6: We would like to thank you for your valuable suggestion. Revisions have been made, and the revised content can be found in Line 167 of the manuscript.

Point 7: Lines 169–170: …image segmentation annotation tool (ISAT)…

Response 7: We would like to thank you for your valuable suggestion. Revisions have been made, and the revised content can be found in Line 183 of the manuscript.

Point 8: Figure 3. The image quality needs improvement.

Response 8: We would like to thank you for your valuable suggestion. Revisions have been made, and the revised content can be found in Line 225 of the manuscript.

Point 9: Lines 267–268: “The detailed architectures of these modules are elaborated in the following subsections.” There are no subsections, so this sentence requires revision.

Response 9: We would like to thank you for your valuable suggestion. Revisions have been made, and the revised content can be found in Line 284 of the manuscript.

Point 10: Lines 278–281: CAM(), ?max(), C1D3×3(), σ() — do these parentheses have any specific meaning?

Response 10: They have no special meaning. They are only used to indicate that the terms are functions when explaining the formulas. Please refer to Lines 294–313 of the manuscript for details.

Point 11: 3.1 Experimental setup and 3.2 “Results analysis” should be placed under Materials and Methods.

Response 11: Thank you for your constructive comment on the structural arrangement of "3.1 Experimental Setup" and "3.2 Results Analysis". We have carefully considered your suggestion to incorporate these two sections into "Materials and Methods" and acknowledge the academic rigor behind it. However, after repeated discussions, we still wish to retain the original section arrangement. Given the significance of both the experimental setup and results analysis, we believe it is necessary for them to form an independent chapter. We sincerely hope you can understand this choice.

Point 12: In Table 1, what do the numbers 1, 2, 3, 4, 5, 6 under SOLOv2 refer to?

Response 12: Thank you for your attention to the meanings of the numbers 1, 2, 3, 4, 5, and 6 under the "SOLOv2" column in Table 1. After review, we recognize that these numbers are likely to cause confusion due to the lack of corresponding explanations, and the originally designed "numerical labeling + subsequent interpretation" mode has the issue of unintuitive information transmission. To present the differences between the various ablation experiment variants of the SOLOv2 baseline model more clearly, we have directly deleted the numerical labels 1–6 under the "SOLOv2" column in Table 1. Please refer to Line 454 of the manuscript for the revised content.

Point 13: Table 2 is overly cluttered and requires adjustment.

Response 13: We would like to thank you for your comment. We have adjusted the format of the table, and the revised content can be found in Line 478 of the manuscript.

Point 14: Table 2 presents different models; where in the overall pipeline are these models used or swapped? I suggest adding a complete model architecture diagram.

Response 14: All models presented in Table 2 are used for performance comparison with the model proposed in this paper. For details, please refer to Lines 456–476 of the manuscript.

Point 15: Are AP, APs, APm, and APl values all greater than 0.7 correct? In general, for AP: >50% (0.50) is very good (SOTA on COCO); 30%–50% is acceptable/practical; <20% is underperforming. APm targets medium-sized objects (32² ≤ area < 96² pixels). It is usually higher than APs—the higher, the better. Typical ranges: 30%–50% is standard; 50% is excellent.

Response 15: Thank you for your professional question regarding the rationality of the four metric values (all greater than 0.7) for AP, APs, APm, and APl, as well as your valuable reference to the typical AP range. We hereby confirm that the values of the aforementioned metrics exceeding 0.7 in the manuscript are statistically valid. All studies in this paper adopt a self-built dataset, and the four metric values of the SOLOv2 baseline model are already relatively high (please refer to Line 478 of the manuscript). The model proposed in this paper further improves its performance and addresses the issues existing in the baseline model.

Point 16: Figure 9b is not explained in the text.

Response 16: Thank you for your valuable comment. We have added an explanation for Figure 9(b) in the manuscript, and the revised content can be found in Line 486.

Point 17: Figure 10: The image arrangement may need to be reworked.

Response 17: Thank you for your valuable comment. We have revised the layout of Figure 10, and the revised content can be found in Line 501 of the manuscript.

Point 18: Figure 11, Lines 482–484: “Figure 11 presents additional segmentation examples. Due to space limitations, only the full-image segmentation results are displayed here, without zoomed-in comparative analysis.” Since there is no accompanying explanation or analysis, its inclusion does not provide meaningful value to the paper. I suggest describing the picture.

Response 18: Thank you for your suggestion. We have added a brief description in the manuscript, which can be found in Lines 506–507.

Point 19: Table 3: How are the confidence levels computed? “Highly confident (>0.9),” “Confident (0.7–0.9),” and “Low confidence (<0.7)” do not match the “%” units used in the content.

Response 19: Thank you for your valuable comment. We have added the statistics of confidence, which can be found in Line 506. Additionally, the "%" refers to the proportion relative to the whole, and this is explained in the manuscript (see Lines 511–517).

Round 2

Reviewer 2 Report

Comments and Suggestions for Authors

The author has responded positively to most comments and made satisfactory revisions in several areas. The manuscript quality has improved, but further clarification and refinement are still needed regarding experimental comparisons. The main flaw is as follows.

Regarding comment 4: The authors' response to the comparison with existing methods is acknowledged. However, even though instance segmentation has been superseded to some extent by panoptic segmentation, the comparison with segmentation methods from the last three years would be more helpful in demonstrating the performance of the proposed method.

Reviewer 3 Report

Comments and Suggestions for Authors

The authors have revised the manuscript and responded to the issues I previously raised. However, I still find that the way references are cited in the Introduction section is somewhat awkward, and I suggest that the authors further refine it to better follow academic conventions and improve readability.

In addition, the caption format of Table 1 should be consistent with the other tables, particularly regarding the use of font size and capitalization.

Finally, for Figure 10, I recommend placing the labels of each sub-figure at the top-left corner, and enclosing each group of images within a bounding box. This would make the figure easier to read and the comparisons clearer.
